# The Inflammasome in Chronic Complications of Diabetes and Related Metabolic Disorders

**DOI:** 10.3390/cells9081812

**Published:** 2020-07-30

**Authors:** Stefano Menini, Carla Iacobini, Martina Vitale, Giuseppe Pugliese

**Affiliations:** Department of Clinical and Molecular Medicine, “La Sapienza” University, 00189 Rome, Italy; stefano.menini@uniroma1.it (S.M.); carla.iacobini@gmail.com (C.I.); vitale.martina1987@gmail.com (M.V.)

**Keywords:** cardiovascular disease, diabetic kidney disease, diabetic retinopathy, non-alcoholic fatty liver disease, NOD-like receptor pyrin domain-containing-3, toll-like receptors

## Abstract

Diabetes mellitus (DM) ranks seventh as a cause of death worldwide. Chronic complications, including cardiovascular, renal, and eye disease, as well as DM-associated non-alcoholic fatty liver disease (NAFLD) account for most of the morbidity and premature mortality in DM. Despite continuous improvements in the management of late complications of DM, significant gaps remain. Therefore, searching for additional strategies to prevent these serious DM-related conditions is of the utmost importance. DM is characterized by a state of low-grade chronic inflammation, which is critical in the progression of complications. Recent clinical trials indicate that targeting the prototypic pro-inflammatory cytokine interleukin-1β (IL-1 β) improves the outcomes of cardiovascular disease, which is the first cause of death in DM patients. Together with IL-18, IL-1β is processed and secreted by the inflammasomes, a class of multiprotein complexes that coordinate inflammatory responses. Several DM-related metabolic factors, including reactive oxygen species, glyco/lipoxidation end products, and cholesterol crystals, have been involved in the pathogenesis of diabetic kidney disease, and diabetic retinopathy, and in the promoting effect of DM on the onset and progression of atherosclerosis and NAFLD. These metabolic factors are also well-established danger signals capable of regulating inflammasome activity. In addition to presenting the current state of knowledge, this review discusses how the mechanistic understanding of inflammasome regulation by metabolic danger signals may hopefully lead to novel therapeutic strategies targeting inflammation for a more effective treatment of diabetic complications.

## 1. Introduction

Diabetes mellitus (DM) is a chronic metabolic disease characterized by hyperglycemia resulting from a defect in insulin secretion, insulin sensitivity, or both. Genetic and environmental factors—the latter associated with lifestyle such as unhealthy diet, physical inactivity/sedentariness, overweight/obesity, and alcohol and tobacco consumption—are well-recognized contributors to the increasing incidence of DM and its complications [1]. DM is a major cause of cardiovascular disease (CVD), such as heart attack and stroke, lower limb amputation, and microvascular disease leading to retinopathy, nephropathy, and neuropathy [2]. Understanding the mechanisms of DM-induced organ damage is critical to prevent and delay long-term complications and to reduce the health and economic burden of DM.

According to the World Health Organization, the prevalence of DM is steadily increasing, particularly in low- and middle-income countries where the percentage of deaths attributable to DM that occurs prior to age 70 is higher than in high-income countries [2]. Globally, an estimated 422 million adults were living with this metabolic disease in 2014 and 1.6 million died as a result of their DM in 2016, with DM ranking the seventh cause of death [2]. Considering that the prevalence of DM is predicted to rise to 642 million by 2040 [3] and that the principal cause of DM mortality is CVD [2,3], improving strategies to prevent long-term complications of DM is more relevant than ever.

Systemic, sterile, low-grade chronic inflammation has long been recognized as a central biological mainstay of DM, and growing evidence suggests that inflammation plays an important role in DM-related complications (Figure 1) [4,5]. Over the recent past, great interest has developed regarding the involvement of the immune system in the development of chronic complications of DM and related metabolic disorders. The aberrant immune cell activation is thought to be one of the contributing mechanisms to the development of DM-related CVD [6,7].

Among the complex network of pro-inflammatory cytokines that are related to chronic metabolic diseases, interleukin (IL)-1β has been recognized to be an important player in initiating and sustaining inflammation-induced organ dysfunction in DM [8,9,10]. Therefore, the identification of factors capable of modulating the secretion of bioactive IL-1β would have therapeutic implications. In this effort, a growing number of studies on the pathogenesis of diabetic complications have focused on the involvement of multi-protein scaffolding complexes, termed inflammasomes, that coordinate the inflammatory response. These components of the innate immune system are signaling platforms that, once assembled, lead to caspase-1 activation and the processing of pro-IL-1β and pro-IL-18 to their mature bioactive forms [11,12], as well as to gasdermin D-mediated pyroptosis [13], an inflammatory form of programmed cell death. Classically, inflammasomes consist of an upstream sensor protein of the NOD-like receptor (NLR) family, the adaptor protein Apoptosis-associated speck-like protein containing a CARD (ASC), and the downstream effector cysteine protease procaspase-1. Several NLR proteins (NLRPs) capable to initiate the formation of an inflammasome were reported, including NLRP1, 2, 3, 6, and 7, NLRC4, and the hematopoietic expression, interferon-inducible nature, and nuclear localization (HIN) domain-containing family members absent in melanoma 2 (AIM2) and interferon γ-inducible protein 16 (IFI16) [12,14].

In this review, we outline the mechanisms of inflammasome activation and regulation in DM and related metabolic diseases and summarize the state of the art of our understanding of the involvement of inflammasome in the pathogenesis of chronic complications of these disorders. Along with this, we also discuss how the mechanistic understanding of inflammasome regulation by metabolic danger signals may hopefully lead to novel therapeutic strategies targeting the inflammasome for a more effective treatment of diabetic complications.

## 2. Activation and Regulation of the Inflammasome in Diabetes Mellitus and Related Metabolic Disorders

In metabolic research, particularly on DM and its chronic complications, much of the attention has been focused on NACHT, LRR, and PYD domains-containing protein 3 (NLRP3), also known as cryopyrin. This is the most characterized and investigated member of the NLR family. NLRs cooperate with toll-like receptors (TLRs) in the innate immune response to pathogens. Functionally, they are pattern recognition receptors (PRRs) that can be activated in response to a wide range of microbial and non-microbial (i.e., sterile) insults [15]. Like TLRs, NLRs are found in cells of the myeloid lineage (i.e., macrophages, granulocytes lymphocytes, etc.) and also in non-immune cells (e.g., endothelial cells) [15,16]. The current view is that NLRP3 activation in macrophages is a two-step process [17] requiring a first signal provided by microbial or endogenous molecules, termed pathogen- (PAMPS) and damage- (DAMPs) associated molecular patterns, respectively. The first signal induces a priming event which triggers NLRP3 and pro-IL-1β expression through activation of nuclear factor (NF)-κB [18,19], though more recent evidence demonstrates that priming also regulates NLRP3 activation at the post-transcriptional level [20,21]. The second signal (activation) is provided by a number of factors, including pore-forming toxins, adenosine triphosphate (ATP), viral RNA, or endogenous crystals and particulate matter [17]. The disparate chemical and structural nature of the NLRP3-activating stimuli makes their direct interaction with NLRP3 unlikely and rather suggests a common cellular signal in response to these NLRP3 activators. However, to date, there is no agreement on the identity of the signal involved in NLRP3 activation and it is not clear whether different activating stimuli can induce different signals. In fact, several cellular and molecular events have been suggested to initiate the assembly and activation of NLRP3, including mitochondrial dysfunction, lysosomal rupture, reactive oxygen species (ROS), K^+^ efflux, and Ca^2+^ signaling [17].

Multiple DM-related metabolic factors may function as NLRP3-inducing stimuli [7,22] (Figure 2). The increased production of ROS [23,24,25,26] and advanced glycation end-products (AGEs) [26,27,28], enhanced lipotoxicity due to high free fatty acids (FFAs) and cholesterol levels [24,29,30,31], defective autophagy and unfolded protein response [23,32], as well as uric acid [33] and extracellular ATP levels [34,35] have been directly involved in the development of diabetic complications and are widely recognized as potent metabolic danger signals able to regulate inflammasome activity [7,24,29,30,35,36,37,38,39].

Depending on the metabolic alteration (i.e., insulin resistance, hyperglycemia, gout, dyslipidemia, etc.) and the tissue (i.e., vascular wall, kidney, liver, etc.) considered, a plethora of metabolic danger signals have been identified so far. Given that different DAMPs have been reported to induce different control mechanisms of inflammasome assembly and activation, including mitochondrial dysfunction, endoplasmic reticulum stress, and lysosomal rupture, [40], the specific DAMP-induced mechanisms will be discussed in the paragraphs regarding each chronic complication of DM.

## 3. Diabetic Cardiovascular Disease

DM is a prime risk factor for CVD, conferring a two- to three-fold greater risk of major adverse cardiovascular events (MACEs), such as heart attack and stroke [2,3]. Together with hypertension, accelerated atherosclerosis and the increased prevalence of chronic kidney disease are the major contributors to DM-associated CVD [32,41]. The classical therapeutic approach for diabetic atherosclerosis and associated MACEs includes strict glycemic, lipid, and blood pressure control [42,43], though intensive blood glucose regimen is effective only if initiated early in the course of the disease [44]. Recent evidence from cardiovascular outcome trials has indicated that two new classes of anti-hyperglycemic agents, i.e., the Glucagon-like peptide (GLP)-1 receptor agonists and the sodium glucose cotransporter (SGLT) 2 inhibitors, are cardioprotective for diabetic patients beyond their glucose-lowering effect [45]. However, despite improvements in CVD preventive strategies, there is a need for novel therapies targeting the biological bases of the -accelerating effect of DM on atherogenesis. These may include blocking inflammation, which is a major mechanism of injury in vascular dysfunction, particularly in DM subjects [8,46]. Consistently, the recent reports from the Canakinumab Anti-Inflammatory Thrombosis Outcomes Study (CANTOS) indicate that IL-1β inhibition with a monoclonal antibody (canakinumab) is effective in preventing MACEs among subjects with and without DM. This clinical outcome was associated with the inhibition of inflammation, as attested by the large reduction in circulating high-sensitivity C-reactive protein (hsCRP) and IL-6 levels [8,47,48]. Interestingly, the beneficial effects of IL-1β antagonism on atherosclerosis occurred despite no effect on low-density lipoprotein cholesterol levels and incident DM, indicating that MACE reduction was not dependent on the improvement of dyslipidemia or prevention of DM onset during the study [8,47,48]. These clinical findings indicate that the benefits of IL-1β antagonism on atherosclerosis are solely due to the inhibition of inflammation, and are in agreement with previous experimental studies demonstrating that IL-1β deficiency and IL-1 receptor antagonism were successful in decreasing the severity of lesions in murine models of atherosclerosis [49,50]. Similar results were also obtained in IL-18 deficient mice [51]. By contrast, a nonspecific approach with the generic anti-inflammatory agent methotrexate did not reduce IL-1β nor lowered MACEs in the recent Cardiovascular Inflammation Reduction Trial (CIRT) [52]. Altogether, these findings support a role for inflammasome activation in the atherosclerotic process, as IL-1β and IL-18 are processed to their mature and bioactive forms by this critical component of the innate immune system.

In a pioneering study, Lee et al. laid the foundation for investigating the role of NLRP3 in DM-associated CVD [7]. Increased expression of the inflammasome components NLRP3 and ASC was found in monocytes from newly identified, untreated type 2 DM subjects, along with increased basal and induced inflammasome activation when exposed to DAMP signals. Consistently, drug-naïve type 2 DM patients had higher IL-1β and IL-18 serum levels compared with healthy subjects, and in vitro knockdown of *NLRP3* in monocytes of DM patients prevented the ability of metabolic DAMPs to induce IL-1β and IL-18 secretion [7]. In the same direction were the results of a preclinical study on a rat model of type 2 DM showing that excessive activation of Nlrp3 and pyroptosis were associated with cardiac structural and functional pathological changes, which were reverted by *Nlrp3* silencing [53]. In cooperation with TLR2, NLRP3 inflammasome activation in cardiac macrophages was also involved in DM-induced, potentially life-threatening arrhythmias, which could be successfully treated by either NLRP3 inflammasome inhibition or IL-1 receptor antagonism [54]. Preliminary genetic data have also suggested an association between polymorphisms in inflammasome coding genes and increased risk for diabetic macrovascular complications, especially myocardial infarction [55].

Mechanistic insights into inflammasome activation in diabetic macrovascular complications come from experimental studies in animal and cellular models of atherosclerosis. NLRP3 was involved in hyperglycemia-induced endothelial inflammation, both in vitro and in vivo. In human umbilical vein endothelial cells and in atherosclerotic plaques of diabetic mice, the overexpression of adhesion molecules induced by high glucose was suppressed by both NLRP3 knockdown and IL-1 receptor antagonism [56]. In addition, carotid atherosclerosis was positively associated with plasma levels of IL-1β in DM subjects [56]. Similarly, another study found that high glucose induced the assembly and activation of NLRP3 inflammasome in rat endothelial cells, an effect that could be reverted by blocking NADPH oxidases 4-dependent ROS generation [57]. In addition to hyperglycemia-induced ROS overproduction, oxidized LDLs (oxLDLs) [58] and high mobility group box Protein 1 [59], two well-known agonist ligands of the receptor of AGEs (RAGE) [60,61], have been involved in the activation of NLRP3 that accompanies the atherosclerotic process. Although AGEs have been definitively involved in diabetic atherosclerosis [27,62], it is currently unknown whether the AGE/RAGE axis contributes to the atherogenic process through the activation of NLRP3 (Figure 3).

Besides glucose toxicity, disrupted lipid metabolism in diabetic vessels might also be involved in inflammasome regulation. Increased aortic levels of sterol regulatory element binding protein (SREBP)-1 were associated with elevated expression of NLRP3 inflammasome components in a porcine model of atherosclerosis and DM [63]. Intense immunostaining for this lipogenic transcription factor was observed in macrophages and endothelial cells of early lesions (i.e., fatty streaks) as well as in the fibrous cap and cholesterol crystals of advanced atherosclerotic lesions [63]. Interestingly, similar findings were observed in aortic biopsy specimens from humans with atherosclerosis and DM [63]. It is now a widely accepted notion that cholesterol and its by-products, such as oxLDLs and cholesterol crystals, are metabolic stress factors playing a causal role in vascular inflammation and atherosclerosis. Among the metabolic danger signals, cholesterol crystals are now regarded as the main trigger for NLRP3 activation in atherosclerosis [64]. Besides inducing lysosomal damage, it is currently under debate whether cholesterol crystals would induce NLRP3 activation through inciting the formation of neutrophil extracellular traps (NETs) [65]. NETs are web-like structures released by neutrophils and composed of DNA, myeloperoxidase, proteases, etc. that were previously implicated in atherothrombosis [66]. However, the mechanism linking NETs to NLRP3 activation has not yet been delineated and warrants future investigation (Figure 3).

Some studies also suggest that NLRP3 activity may be modulated by mechanical forces [16,64], including hemodynamic stress [16], that can modify endothelial cell response patterns. Oscillatory shear stress was identified as a novel regulator of endothelial inflammasome activation for atherogenesis through forkhead box P transcription factor 1 (Foxp1) downregulation via repressing Kruppel-like factor 2 expression in endothelium [16]. Consistently, Foxp1 expression was significantly reduced in human and mouse coronary atherosclerotic endothelium [16]. Moreover, mitochondrial DNA was recently shown to contribute to DM-associated endothelial dysfunction and vascular inflammation via NLRP3 activation through mechanisms that involve Ca^2+^ influx and ROS generation [67].

A final note concerns the possible role of the anti-inflammatory effects of the new anti-diabetic medications SGLT2 inhibitors in providing their cardiovascular benefit in type 2 DM patients [68]. The recent finding that SGLT2 inhibitors attenuate NLRP3 inflammasome activation [69] might help to explain the cardioprotective effects of these anti-diabetic drugs. Therefore, this is an area that deserves further investigation.

## 4. Diabetic Kidney Disease

Diabetic kidney disease (DKD) is the leading cause of chronic kidney disease and end-stage kidney disease (ESKD) in most developed countries [70,71]. Approximately 30% of patients with type 1 DM and approximately 40% of patients with type 2 DM eventually develop this microvascular complication [71,72]. DKD patients, even in the early disease stages, carry a significant financial burden to the national health systems due to both direct (e.g., laboratory tests, medications, dialysis, etc.) and indirect (e.g., increased CVD morbidity) costs [73]. In the classical view of the natural history of DKD, five stages are recognized according to changes in urinary albumin excretion rate and glomerular filtration rate (GFR): hyperfiltration (stage 1), normoalbuminuria or intermittent episodes of microalbuminuria (silent nephropathy, stage 2), persistent microalbuminuria (incipient nephropathy, stage 3), macroalbuminuria and progressive GFR reduction (overt nephropathy, stage 4), and, finally, ESKD (stage 5) [74]. Recently, the traditional view that albuminuria invariably precedes and promotes GFR loss has been challenged by increasing evidence suggesting that both initiation and progression of GFR decline may occur also independently of albuminuria or even in the absence of it [75,76]. At least in part, such changes in the clinical course of DKD are likely a consequence of the introduction into clinical practice of blockers of the renin-angiotensin system (RAS), the increasing use of which has not translated into a lower prevalence of reduced GFR and DKD, and has only marginally decreased the incidence of ESKD [76]. Moreover, it is currently unclear if the heterogeneity in the clinical presentation and course of DKD reflects different anatomic entities and pathogenic mechanisms.

The current management scheme for DKD includes strict glycemic control, anti-hypertensive drugs, lipid-lowering treatment, obesity reduction, and dietary and lifestyle adjustments. DKD has been traditionally considered as the result of the interaction between hemodynamic factors, such as hyperfiltration, hypertension, and persistent local RAS activation, and metabolic factors, such as hyperglycemia and dyslipidemia. Recent evidence also indicates an important role for immune and inflammatory mechanisms, which have been implicated in the development of DKD and its progression to ESRD [77,78]. However, the inflammatory mediators, their sources (i.e., systemic or renal), and the molecular mechanisms underlying the relationship between chronic inflammation and DKD are largely unknown. Using an aptamer-based proteomic approach to examine 194 circulating inflammatory proteins in subjects with type 1 and type 2 DM, a circulating kidney risk inflammatory signature (KRIS) composed of 17 inflammatory proteins has been recently identified [79]. In addition to tumor necrosis factor (TNF) receptors 1 and 2, which were previously identified as robust predictors of renal function decline in type 1 and type 2 DM [80], 15 new circulating inflammatory proteins, including other four members of the TNF-receptor superfamily, were found to be strongly associated with renal outcomes in people with DM. Of note, all these inflammatory proteins had a systemic, non-renal source. Therefore, the signature comprising KRIS proteins highlights the importance of systemic factors as relevant pathogenic factors in DKD.

One plausible non-renal source of these inflammatory proteins are leucocytes. However, renal monocyte infiltration is observed in the course of both human and experimental DKD [81,82], and the inhibition of macrophage/monocyte recruitment into the kidney has been shown to be protective in experimental DKD models [78]. Therefore, targeted anti-inflammatory therapy has been suggested for both prevention and treatment of DKD [83].

Activated macrophages are known to secrete large amounts of cytotoxic products, including both pro-inflammatory and pro-fibrotic cytokines that may contribute to renal cells dysfunction and tissue injury. Among pro-inflammatory cytokines, serum and urinary levels of IL-18 and IL-1β have been reported to be higher in patients with DM [84,85]. While IL-1β secretion is largely restricted to infiltrating monocytes, IL-18 is also expressed by kidney cells [86,87], particularly upon kidney injury [86]. Accordingly, IL-18 has been proposed as a useful urinary biomarker of kidney injury [87]. However, essential elements of the NLRP3 inflammasome were also found to be expressed in podocytes [88,89], renal endothelial cells [88,90] and tubular epithelial cells [91,92,93,94,95]. Consistently, pro-IL-1β processing and IL-1β release from renal parenchymal cells has been reported [9,88]. Moreover, ongoing renal tissue injury exposes renal cells to numerous inflammasome activators, such as uric acid, ATP, uromodulin, histones, oxalate or cystine crystals, and matrix degradation products [39,87,96,97,98,99]. Altogether, these findings suggest that inflammasomes may contribute to the pathogenesis of kidney damage through the production of pro-inflammatory cytokines by both infiltrating immune cells and non-immune, resident renal cells [80].

Significant evidence exists supporting the importance of caspase-1-dependent NLRP3 inflammasome activation in the pathogenesis of DKD. Circulating IL-1β and IL-18 levels, glomerular expression of IL-1β and IL-18 and inflammasome markers (NLRP3, ASC, etc.) were found to be increased in db/db mice, an experimental model of type 2 DM. Importantly, all these increases preceded structural and functional abnormalities characteristic of diabetic glomerulopathy, such as extracellular matrix accumulation and albuminuria, suggesting that NLRP3 activation triggers the onset of DKD. Importantly, all these changes were significantly reduced by NLRP3 or caspase-1 deficiency, and by pharmacological inhibition of inflammasome or caspase-1 activation [9,88]. Moreover, the transplantation of Nlrp3- or caspase-1-deficient bone marrow (BM) cells failed to ameliorate albuminuria and mesangial expansion in wild type diabetic mice, whereas *Nlrp3*^-/-^ diabetic mice maintained protection from DKD despite transplantation of wild type BM cells, indicating that the Nlrp3 inflammasome in renal resident cells, particularly podocytes and glomerular endothelial cells, is the main contributor to the pathogenesis of DKD [9] (Figure 4). Finally, renal inflammasome activation was also observed in DM patients, in whom glomerular NLRP3 expression was found to be increased compared with non-diabetic subjects. In addition, serum IL1β levels were found to be higher in diabetic than in non-diabetic subjects and, among DM individuals, in albuminuric than in non-albuminuric patients [9].

A number of experimental studies have subsequently provided additional mechanistic information about the involvement of the NLRP3 inflammasome in DKD. Among the mechanisms implicated, glyco- and lipotoxicity were shown to promote inflammasome activation in the diabetic kidney through the mitochondrial ROS/Thioredoxin Interacting Protein (TXNIP) pathway [100,101,102]. In addition, the long intergenic noncoding RNA-Gm4419 was found to activate the NF-κB pathway/NLRP3 inflammasome signaling in glomerular mesangial cells exposed to high glucose conditions [103]. Recently, there has been an increasing interest in the ncRNA field and, specifically, in the role of ncRNAs in chronic complications of diabetes and related metabolic disorders. However, this topic has been reviewed recently elsewhere [104,105,106]. Mitochondrial ROS [9], uric acid [96], and ATP signaling through the purinergic receptor 2X7 (PR2X7) [34,107] have also been involved in the activation of the NLRP3 inflammasome in the diabetic kidney. Finally, D-ribose, the level of which is increased in DM, was demonstrated to induce NLRP3 inflammasome assembly and activation in podocytes via AGEs/RAGE signaling pathway [108].

Among current anti-diabetic drugs, dipeptidyl peptidase (DPP) 4 inhibitors [109], SGLT2 inhibitors [109,110], and thiazolidinediones [111] have been reported to have pleiotropic anti-inflammatory activity by reducing NLRP3 activation in DKD experimental models, independently of their glucose lowering effect. These are preliminary results obtained in preliminary experimental studies. Nonetheless, they support the need for more strictly controlled studies aimed at searching for new agents specifically targeting the NLRP3 inflammasome as potential therapeutic tools for DKD. Currently, there are only a few studies on this topic, investigating the anti-inflammasome activity of minocycline [112], curcumin [113], IL-22 [114], IL6-receptor blockade [115], and thrombomodulin gene therapy [116], with suggestive but not entirely conclusive results. To complicate matters, a protective role of the NLRP1 inflammasome in DKD was recently suggested [117]. It follows that there remains a need for more determined steps in the direction of understanding the clinical relevance of inflammasome signaling in DKD.

## 5. Diabetic Retinopathy

Diabetic retinopathy (DR) is a common complication of type 1 and type 2 DM, and a major cause of visual impairment and blindness among working-age and elderly people [118]. Nearly 35% of people with DM have DR and the prevalence increases with age and disease duration [118,119]. DR is a multifactorial disease with a complex interaction of microvascular, genetic, neurodegenerative, immunological, and inflammatory-related events. DR can be subdivided into two stages: the earlier stage of non-proliferative DR (non-PDR) and the advanced stage of PDR. Non-PDR lesions include microaneurysms, retinal hemorrhages, changes in vascular caliber and capillary nonperfusion [120], while PDR is characterized by pathological preretinal neovascularization coupled with fibrosis at the vitreoretinal interface [121].

By producing pro-inflammatory and pro-angiogenic cytokines, macrophages are actively involved in an aberrant ocular healing response. Persistent tissue inflammation leads to chronic tissue damage and repair resulting in severe fibrosis (gliosis) and blood-retinal barrier breakdown, with consequent extravasation of fluid into the neural retina [122,123,124]. A leakage of fluid and circulating lipoproteins from retinal vessels results in the formation of hard exudates and macular edema, a severe complication of DR that can occur both in non-PDR and PDR, and nowadays represents the leading cause of blindness in patients with DM [120,121]. 

Many factors may influence DR, including hyperglycemia, dyslipidemia and hypertension, but the underlying biochemical and molecular mechanisms are still poorly defined. Seminal large-scale clinical trials have provided evidence that hyperglycemia is closely linked to DR, as intensive glycemic control delayed the onset and slowed the progression of this microvascular complication in both type 1 [42] and type 2 [43] DM. The relationship between hyperglycemia and chronic low-grade inflammation is believed to play a role also in DR. However, it is not completely clear how inflammation is initiated in DR and whether it is the initial instigator or an adjunct player in the pathogenic process. Microglia and endothelial cell activation may play a role in early vascular and neural inflammation in DR [125,126,127]. In the initial stages of DR, pro-inflammatory molecules produced by both microglia and endothelial cells may in fact be responsible for retinal leukostasis and leukocyte extravasation (Figure 5). This view is supported by the finding that chronic hyperglycemia induced the upregulation of the oxidative sensor and NLRP3 modulator TXNIP, with consequent activation of inflammatory signaling pathways in both glial and microvascular endothelial cells [126,128]. As DR progresses, multiple immune cell types, including neutrophils and monocytes, accumulate in the retina, leading the local concentration of inflammatory molecules to rise [121,125]. Consistently, increased intravitreal levels of proinflammatory cytokines including IL-6, TNF-α, Monocyte Chemoattractant Protein 1, hsCRP, and the inflammasome-related cytokines IL-1β and IL-18, have been detected in DR patients [129,130,131,132]. This adds to the growing body of evidence that inflammatory pathways, and possibly inflammasome activation, play a crucial role in DR development. 

In the last decade, and particularly over the last four years, experimental studies have provided proof of the involvement of various members of the inflammasome family in DR. A study performed in the high-fat fed model of obesity-related type 2 DM demonstrated an association between inflammasome activation, attested by NLRP3 upregulation and increased levels of cleaved IL-1β and caspase-1, and retinal dysfunction. What is more, NLRP3 activation and electroretinographic defects preceded the appearance of microvascular disease, providing the rational for targeting the NLRP3 inflammasome to treat DR [133]. These findings are consistent with data from humans, where functional deficits are clinically detectable before microvascular structural lesions [133,134]. Another study that analyzed the retinas from db/db mice and human diabetic donors to investigate the role of microglia polarization dynamics in DR found that the modulation of this process towards a M2 phenotype at the early stages of DR might have therapeutic relevance [135]. Consistent with a pathogenic role of the NLRP3 inflammasome in human DR, treatment of human retinal endothelial cells with MCC950, a potent and specific inhibitor of the NLRP3 inflammasome, inhibited high-glucose-induced cell dysfunction, including apoptosis [127]. Moreover, elevated NLRP3, caspase 1 and IL-1β expression in proliferative membranes [127], and high levels of IL-1 β and IL-18 in vitreous humor [136] were observed in PDR compared with non-PDR patients. Consistent with a role of NLRP3 inflammasome in the pathological neovascularization of the retina, increased mRNA and protein levels of inflammasome components were also detected in the retina of Akimba mice, a double transgenic model displaying characteristic features of the advanced stages of DR (i.e., PDR) [137]. Finally, the expression of NLRP1, another member of the NLR family, was recently found to be increased in streptozotocin (STZ)-induced diabetic mice with features of DR [138], but not in the Akimba mouse model of PDR [137]. The finding that DR was attenuated in diabetic *Nlrp1-*deficient mice support the view of a contributing role of this NLR in DR [138].

Multiple non-mutually exclusive mechanisms of inflammasome activation in DR have been proposed. Hyperglycemia-induced ROS production may stimulate NLRP3 inflammasome activity through different pathways, including thioredoxin (TRX1)-TXNIP dissociation and accumulation of AGEs and advanced lipoxidation endproducts (ALEs) [139]. ROS-dependent TRX1-TXNIP dissociation was found to stimulate TXNIP-NLRP3 interaction and release of IL-1β in retinal Müller glia under chronic hyperglycemia [126,128]. AGE/ALEs were reported to induce NLRP3 mRNA expression and production of IL-1β and IL-18 in retinal pigment epithelial cells (RPE) cells [140]. Moreover, ROS-dependent TXNIP–NLRP3 association was also observed in macrophages treated with monosodium urate [141], a purine metabolite whose contributing role in DM-induced retinal inflammation was recently demonstrated [142]. Finally, ATP signaling through PR2X7) was shown to contribute to both the hyperglycemia-induced release of IL-1β from retinal pericytes and pericyte loss [143], leading to vascular leakage and macular edema.

Current therapies focus on advanced disease to prevent progression of PDR and vision loss [120]. Standard treatment options for DR include laser photocoagulation, intravitreal injections of anti-VEGF and steroid agents, and vitreoretinal surgery. The only interventions aimed at preventing DR are tight glucose and blood pressure control and much remains to be done in this direction. Most of the known inhibitors of inflammasomes are yet to be tested in humans or animal models of DR. As for DKD, the tetracycline antibiotic minocycline showed potential for treatment benefit in rodent DR, mainly by downregulating the ROS/TXNIP/NLRP3 inflammasome pathway [126]. Similar results were obtained in diabetic rats treated with 1,25-dihydroxyvitamin D [144]. Recent experimental evidence identified Prostaglandin E_2_ (PGE_2_) and its E-prostanoid receptor (EP_2_R) as critical regulators of inflammation and retinal endothelial cell damage in DR [145]. In STZ-induced diabetic rats, the intravitreal injection of PGE_2_ or butaprost, a PGE_2_/EP_2_R agonist, accelerated retinal vascular leakage, leukostasis, and endothelial cell apoptosis. These alterations were attenuated in rats that were pre-treated with AH6809, an EP_2_R antagonist. In addition, pretreatment of human retinal endothelial cells with AH6809 significantly inhibited PGE_2_- and butaprost-induced activation of caspase 1, NLRP3, and ASC [145], suggesting that both PGE_2_/EP_2_R signaling pathway and the inflammasome may be considered as potential target for DR prevention and treatment.

Finally, therapeutic agents commonly used for metabolic disorders may also be taken into consideration for their off-label beneficial effect on DR. An intriguing candidate protective factor for DR is the triglyceride-lowering drug fenofibrate, a peroxisome proliferator-activated receptor-α (PPARα) agonist. Fenofibrate has been demonstrated to delay progression and improve multiple abnormalities in experimental DR, including inflammation and vascular leakage [146,147]. What is more, the benefit was independent of the lipid-lowering activity of this drug [148]. Two large clinical trials showed the efficacy of this PPARα agonist in slowing the progression of DR and reducing retinal laser requirement, [149,150]. Interestingly, recent experimental evidence has been provided that fenofibrate ameliorates DR by modulating nuclear factor erythroid-2-related factor 2 signaling and NLRP3 inflammasome activation [151].

## 6. Diabetic Neuropathy

Nerve injury is a common complication of DM that leads to chronic pain, numbness, weakness and substantial loss of quality of life. Good glycemic control can prevent or retard the development of diabetic neuropathy, but more than half of all patients with diabetes eventually develop this complication [152]. Currently, there is no approved treatment to prevent or treat diabetic neuropathy, and only symptomatic pain therapies of variable efficacy are available. Though traditionally counted among microvascular complications, abnormalities in neuronal cells likely provide a critical contribution to the development of diabetic neuropathy [32]. Unfortunately, the study of the molecular mechanisms and, specifically, pathogenic mechanisms involving the inflammasome has lagged somewhat behind other complications of diabetes. To date, a handful of articles on the involvement of NLRP3 in some pathophysiological changes and clinical manifestations of diabetic neuropathy are found in the literature [153,154,155,156,157]. Though suggesting a contribution of the TXNIP [153,154], PR2X4 [155], and HMGB1/TLR4-NLRP3 signaling pathways [156], these studies do not provide conclusive evidence on the role of NLRP3 in diabetic neuropathy and need to be confirmed in larger controlled studies.

## 7. Diabetes-Associated Non-Alcoholic Fatty Liver Disease

Nonalcoholic fatty liver disease (NAFLD) is one of the most common forms of chronic liver disease, affecting one fourth of the population worldwide [158]. Prevalence estimates even increase among people with obesity and type 2 DM, reaching nearly 75% of the individuals affected by DM [158,159]. NAFLD is defined as the presence of ≥5–10% of hepatocytes containing lipid droplets after the exclusion of other causes of fat accumulation, such as excessive alcohol intake, viral infections, and use of steatogenic medications. NAFLD encompasses a wide spectrum of liver pathology, from the simple accumulation of fat (hepatic steatosis) to nonalcoholic steatohepatitis (NASH), cirrhosis, and liver cancer [160]. Hepatic steatosis is a reversible condition characterized by hepatic fat accumulation with no evidence of inflammation and hepatocellular injury [158,159,160]. It represents the hepatic manifestation of the metabolic syndrome [161], a condition closely associated with insulin resistance [162]. By favoring FFA mobilization from adipose tissue and increased hepatic FFA uptake and synthesis, insulin resistance favors build-up of extra fat in the liver [163]. Depending on the presence of other factors that act synergistically with insulin resistance, including nutritional and genetic factors, a minority of affected patients develops NASH [164]. Besides steatosis, NASH includes the presence of inflammation with hepatocyte injury, which is often, but not always, associated with fibrosis. Over time, chronic cycles of injury and inflammation may lead to fibrosis progression that can eventually develop into cirrhosis with its clinical consequences [160].

Diabetic patients are at increased risk not only of developing NAFLD [159], but also of progressing to end-stage liver disease. It has been estimated that DM increases the risk of NAFLD progression to NASH by two- to three-fold [165]. In turn, NAFLD confers a higher risk of CVD and malignancies in DM subjects [159]. In this scenario, the understanding of the molecular mechanisms underlying the accelerating effect of DM on the initiation and progression of NAFLD is of extreme importance. Regardless of the presence of DM, both local and systemic stimuli concur in activating and sustaining an inflammatory response in the liver. In the context of the dysregulated glucose metabolism associated with insulin resistance, several DM-related factors have now been recognized to drive progression to NASH and more advanced stages of NAFLD (Figure 6) [166]. These factors include systemic low-grade chronic inflammation, excessive circulating levels and liver uptake of FFAs, intrahepatic de novo lipid synthesis, systemic and liver glucose metabolism alterations, oxidative stress, and the consequent accumulation of ALEs. These are lipotoxic compounds that cause hepatocellular damage and apoptosis, endoplasmic reticulum stress, liver inflammation and fibrosis [167,168]. In addition to intrahepatic formation, circulating ALEs and their glyco-equivalents AGEs, the levels of which are increased in DM [169,170], might accumulate in the liver also because of increased hepatic uptake [171]. Besides the glycotoxic and lipotoxic effects exerted by AGEs and ALEs in the liver, other extrahepatic factors may favor hepatic inflammation in DM patients with NAFLD. Among them, the inflammatory mediators released by dysfunctional adipose tissue, such as IL-6 and TNFα [172], and endotoxins derived from gut microbiota [173,174] have been shown to activate hepatic Kupffer cells to produce IL-1β, TNF-α, and IL-6, thus exacerbating liver inflammation, hepatocyte injury, and fibrosis [175]. The search for a link between glyco/lipotoxicity and NASH has received much attention in recent years. In this context, the finding that the expression of NLRP3 and inflammasome-associated cytokines (pro-IL-1β and pro-IL-18) is higher in patients with NASH compared to those with simple steatosis [176] has spurred research on this topic.

Studies performed in *Nlrp3* knockout mouse models or using NLRP3 inhibitors actually suggest that the activation of the inflammasome is important in NAFLD progression to NASH and fibrosis development [176]. Several NAFLD-associated factors have been shown to be effective stimuli for inflammasome activation, including FFAs [177], cholesterol crystals [178], increased ROS production by dysfunctional mitochondria [176,178], and AGE/ALE-mediated toxicity [169,179,180]. In addition, by providing the priming signal, membrane-bound receptors such as the PRRs TLR9 [181] and TLR4 [178] and PR2X7 [107] have been demonstrated to be critical for NLRP3 activation, IL-1β production, and NASH development. As in the kidney and the eye, both BM-derived and resident liver cells, including hepatocytes and hepatic stellate and sinusoidal cells, seem to contribute to AIM2 and NLRP3 inflammasome activation in NAFLD progression. The activation of these two inflammasomes in dietary NASH was found to be mediated by the myeloid differentiation primary response gene 88 and, in the case of AIM2, supported by TLR9 binding activity [182].

In the specific case of NAFLD associated with DM, there is experimental evidence suggesting that ROS-mediated mechanisms play a major role in the activation of NLRP3 inflammasome. In fact, quercetin and allopurinol, two compounds with proven antioxidant properties [183,184], were shown to be effective in reducing both liver inflammation and lipid accumulation under hyperglycemic conditions [185]. Treatment with these compounds significantly reduced liver inflammation via inhibiting the ROS/TXNIP/NLRP3 pathway, and positively regulated lipid liver metabolism by preventing the upregulation of PPARα and the down-regulation of SREBP1c and 2, fatty acid synthase, and liver X receptor α. Accordingly, targeting the ROS-TXNIP-NLRP3 pathway has been proposed as a therapeutic opportunity for preventing the progression of NAFLD to NASH in DM [185]. However, these experimental data should be confirmed by further studies with larger sample size and under genetically and environmentally controlled conditions, as there is conflicting evidence on the role of inflammasomes in NAFLD. In particular, the crosstalk between the gut and the liver should be explored in depth [168], especially in DM. In fact, some evidence indicates that genetic deficiency of the inflammasomes NLRP3 and NLRP6 promotes NAFLD/NASH progression by inducing intestinal microbiota dysbiosis and liver accumulation of abnormal bacterial products via portal circulation [186]. In addition, NLRP6 and NLRP3 ablation has been associated with a worsening of multiple aspects of the metabolic syndrome, including obesity, via modulation of gut microbiota. Accordingly, a protective role in NAFLD has been proposed for NLRP3 and NLRP6. [186].

There is no specific treatment yet for NAFLD irrespective of the presence or absence of DM. One of the possible strategies is to mitigate the metabolic instigators of NASH. Consistently, current therapies targeted at insulin resistance show some benefit in treating NAFLD [187,188]. Insulin-sensitizing anti-diabetic medications, such as metformin and thiazolidinediones, have been proven to be effective in treating NAFLD in DM patients. Another class of anti-diabetic agents, the GLP-1 receptor agonists, was recently demonstrated to improve all the features of NAFLD, including steatosis, inflammation, and fibrosis [189]. Besides its insulinotropic action, additional metabolic effects, including insulin sensitization properties, likely contribute to GLP-1 anti-NASH activity [190]. However, these medications mainly target organs than the liver, especially muscle and adipose tissue. The intrahepatic drug targets include lipid and glucose homeostasis regulators, mitochondrial and antioxidative stress targets in hepatocytes, inflammatory signals in hepatocytes, and pathways related to stellate cell activation and fibrogenesis [158]. The drugs for these targets are currently tested in phase 2 or 3 clinical trials and were recently thoroughly reviewed by Friedman et al. [158]. At the time of writing, there are no clinical trials specifically designed to evaluate the effectiveness and safety of targeting the inflammasomes in NASH. It must be concluded that much experimental work still needs to be done to clarify the role of inflammasomes in NAFLD before moving into clinical studies.

## 8. Conclusions and Perspectives

Inflammation plays a pivotal role in the development of chronic complications of DM, both microvascular and macrovascular, and also in DM-related metabolic disorders such as NAFLD. The clinical and experimental data discussed in this review highlight the role of systemic and local chronic low-grade inflammation as a key etiological process leading to chronic complications of DM. Accordingly, the benefit of directly targeting inflammation in CVD, especially when associated with DM, has been recently confirmed in randomized controlled trials [8,49,50]. Moreover, the benefit provided by current anti-diabetic medications via pleiotropic effects on inflammation further support the concept that modulating the inflammatory process in DM may be a viable option for the prevention and treatment of chronic complications.

Among the anti-diabetic drugs showing a beneficial off-target effect on inflammation, DPP-4 inhibitors, GLP-1 receptor agonists, SGLT2 inhibitors, and thiazolidinediones have been demonstrated to attenuate NLRP3 activation in experimental models of DM, independently of their glucose lowering effect. This observation would suggest the utility of specific inhibitors of inflammasomes as potential additional treatment for DM-related complications. However, current evidence on the potential benefits of such agents for chronic complications of DM and of other metabolic diseases, is limited by several factors, including conflicting findings and gaps that still exist in literature. Among the available therapeutics that target the inflammasomes, only the NLRP3 inhibitor MCC950 has been tested with encouraging results in preclinical studies of DKD [191], DR [127], and NAFLD [192]. The same compound also showed some benefit in atherosclerosis [193], but in the absence of DM. Finally, in this field, current scientific efforts are also directed at characterizing the effects of natural compounds as potential therapeutic strategies for inflammasome-related metabolic disorders. This topic has not been discussed in this review. For recent developments in the research on phytochemicals as regulators of NLRP3 activity in experimental models of DM, please refer to the recent review of Pellegrini et al. [194].

In summary, more strictly controlled experimental studies aimed at searching for and testing new agents specifically targeting the inflammasomes in diabetic complications should be performed before moving forward to clinical testing in humans.

## Figures and Tables

**Figure 1 cells-09-01812-f001:**
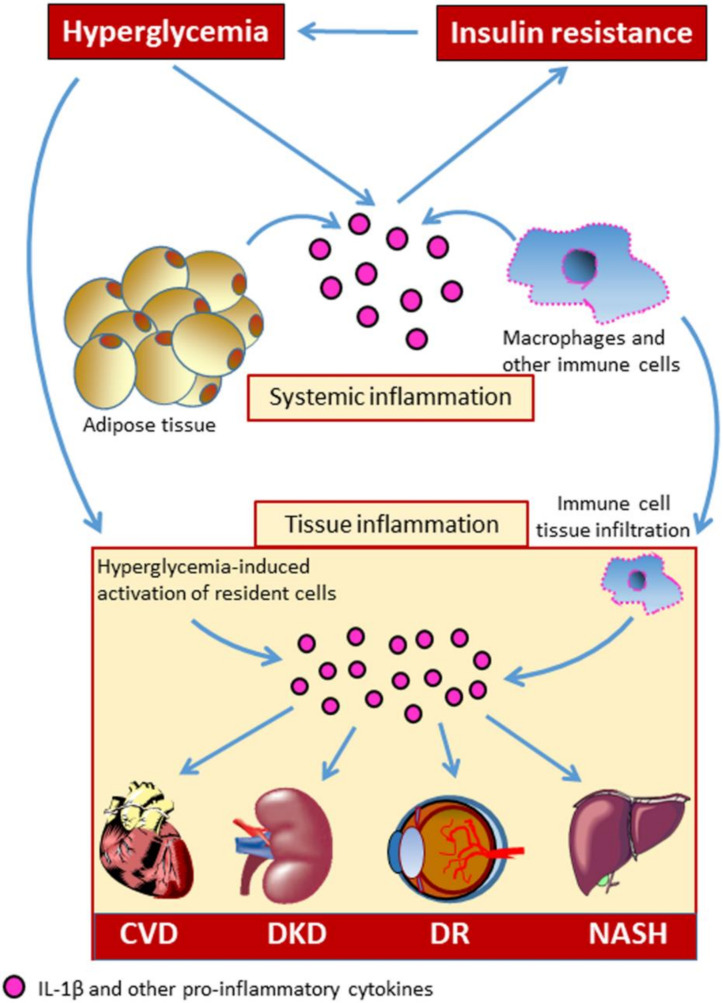
Systemic and local tissue inflammation in diabetic complications. CVD = cardiovascular disease; DKD = diabetic kidney disease; DR = diabetic retinopathy; NASH = non-alcoholic steatohepatitis. IL-1β = interleukin-1β.

**Figure 2 cells-09-01812-f002:**
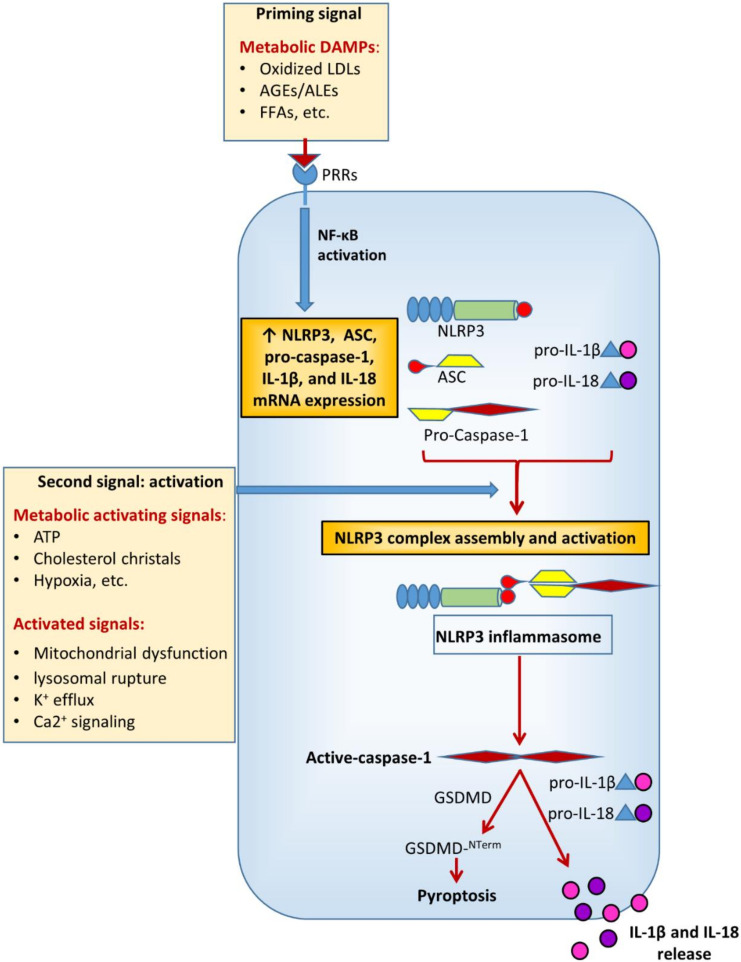
Diabetes mellitus-related metabolic factors acting as NLRP3-priming and activating signals. LDLs = low-density lipoproteins; AGEs = advanced glycation endproducts; ALEs = advanced lipoxidation endproducts; FFAs = free fatty acids; ATP = adenosine triphosphate; ASC = apoptosis-associated speck-like protein containing a CARD; IL-1β = interleukin-1β; IL-18 = interleukin-18; GSDMD = gasdermin D; GSDMD-N = GSDMD N-terimanl domain.

**Figure 3 cells-09-01812-f003:**
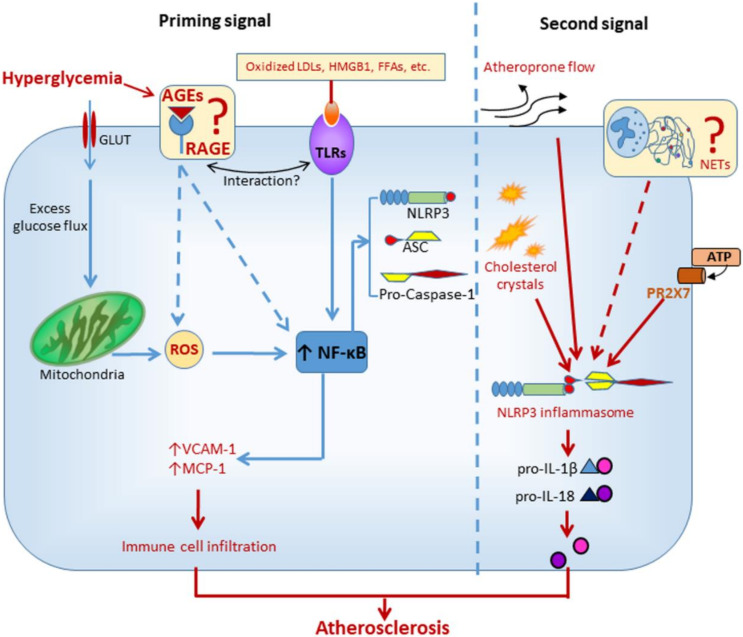
Diabetes mellitus-related factors possibly involved in the activation of NLRP3 and acceleration of the atherogenic process. GLUT = glucose transporter; AGEs = advanced glycation endproducts; RAGE = Receptor for AGEs; TLRs = toll-like receptors; LDLs = loe density lipoproteins; HMGB1 = high mobility group box 1; NETs = neutrophil extracellular traps; ROS = radical oxygen species; NF-κB = nuclear factor kappa-light-chain-enhancer of activated B cells; ASC = apoptosis-associated speck-like protein containing a CARD; P2RX7 = P2X purinoreceptor 7; VCAM-1 = vascular cell adhesion molecule 1; MCP-1 = monocyte chemoattractant protein 1; IL-1β = interleukin-1β; IL-18 = interleukin-18.

**Figure 4 cells-09-01812-f004:**
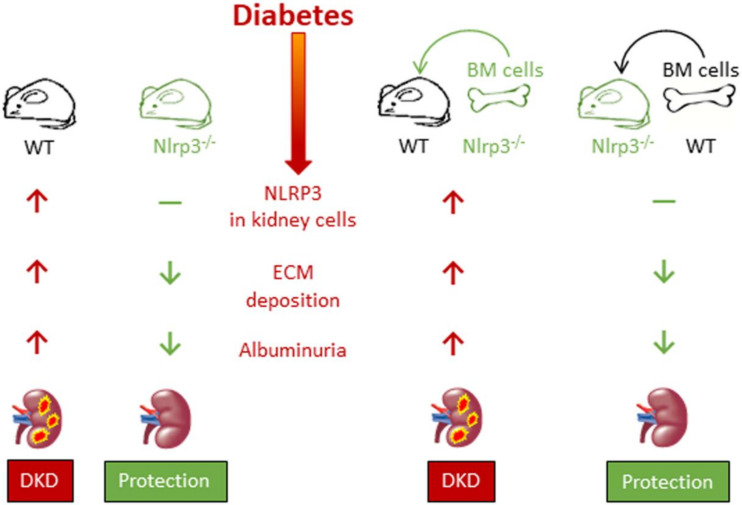
Nlrp3 inflammasome activation in renal resident cells contributes to the pathogenesis of diabetic kidney disease. WT = wild type; NLRP3^−/−^ = NOD-like receptor pyrin domain-containing-3 knocked down; BM = bone marrow; ECM = extracellular matrix; DKD = diabetic kidney disease.

**Figure 5 cells-09-01812-f005:**
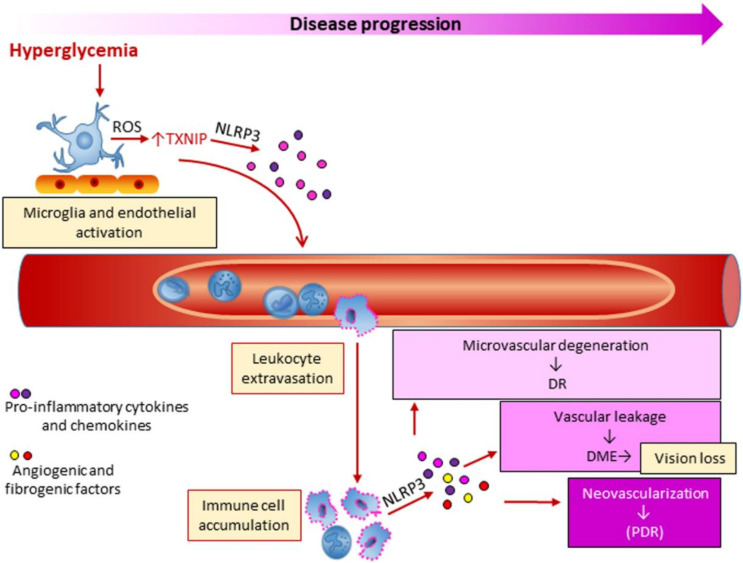
Role of inflammation and inflammasome NLRP3 activation in the early and advanced stages of diabetic retinopathy. ROS = radical oxygen species; TXNIP = thioredoxin interacting protein; DR = diabetic retinopathy; DME = diabetic macular edema; PDR = proliferative DR.

**Figure 6 cells-09-01812-f006:**
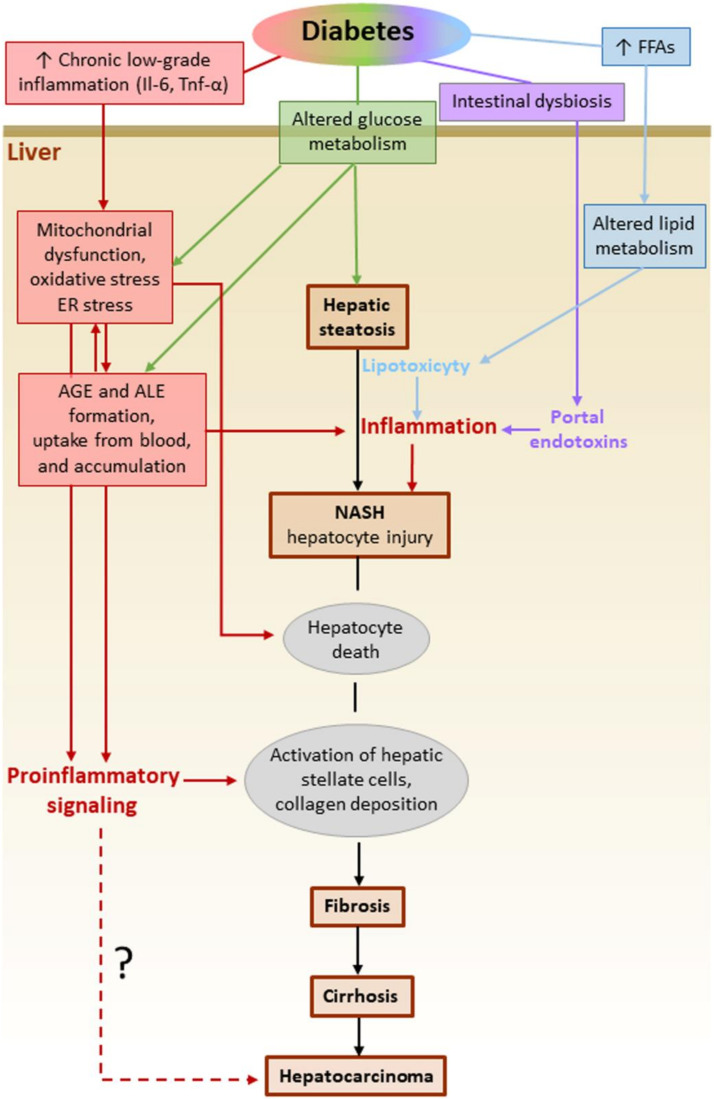
Molecular mechanisms underlying the accelerating effect of diabetes mellitus on the initiation and progression of NAFLD. The dashed line indicates a hypothesis under investigation. IL-6 = interleukin-6; TNF-α = Tumor necrosis factor-α; FFAs = free fatty acids; ER = endoplasmic reticulum; NASH = non-alcoholic steatohepatitis; AGE = advanced glycation endproducts; ALE = advanced lipoxidation endproducts.

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
