# Peer review of "The Inflammasome in Chronic Complications of Diabetes and Related Metabolic Disorders"

_cells, 2020, doi:10.3390/cells9081812_

Round 1

Reviewer 1 Report

The Article by Stefano Menini et al., reviewing the literature on role of inflammation especially inflammasome activation in diabetes and related metabolic disorder. It is one of good review article in the field of DM compiling available literature on inflammasome and outline the mechanisms of inflammasome activation and regulation in DM and related metabolic diseases. It summarize available scientific literature understanding involvement of inflammasome in the pathogenesis of chronic complications diabetes. Authors Further discuss some of the metabolic danger signals produced during DM can regulate inflammasome signaling pathway. Author finally concluded that targeting the inflammasome activation can be more effective treatment option of diabetic complications along with available treatment. However, a few additions could further strengthen the manuscript.

  1. Figure 2 is little confusing and suggest revising or simplifying it.
  2. As author stated in the introduction neuropathy is also one of the diabetic complications along with retinopathy and nephropathy. However, role of inflammasome in diabetic neuropathy is not discussed in the manuscript. Addition of small para compiling available literature on this topic further strengthen the article. This isn't a major concern - just a thought for consideration.

Author Response

Reviewer #1

The Article by Stefano Menini et al., reviewing the literature on role of inflammation especially inflammasome activation in diabetes and related metabolic disorder. It is one of good review article in the field of DM compiling available literature on inflammasome and outline the mechanisms of inflammasome activation and regulation in DM and related metabolic diseases. It summarize available scientific literature understanding involvement of inflammasome in the pathogenesis of chronic complications diabetes. Authors Further discuss some of the metabolic danger signals produced during DM can regulate inflammasome signaling pathway. Author finally concluded that targeting the inflammasome activation can be more effective treatment option of diabetic complications along with available treatment. However, a few additions could further strengthen the manuscript.

We thank the reviewer for her/his overall positive judgement and valuable suggestions and feedback in improving the manuscript

  1. Figure 2 is little confusing and suggest revising or simplifying it.

As requested by the Reviewer, we revised Figure 2.

  1. As author stated in the introduction neuropathy is also one of the diabetic complications along with retinopathy and nephropathy. However, role of inflammasome in diabetic neuropathy is not discussed in the manuscript. Addition of small para compiling available literature on this topic further strengthen the article. This isn't a major concern - just a thought for consideration.

Following to the Reviewer’s suggestion, we have added a short paragraph compiling available literature on the role of inflammasome in diabetic neuropathy.

Reviewer 2 Report

The current manuscript is well written and informative. The authors clearly described the function and signal transduction of inflammasomes in diabetes mellitus. There are no major problems.

Author Response

Reviewer #2

The current manuscript is well written and informative. The authors clearly described the function and signal transduction of inflammasomes in diabetes mellitus. There are no major problems.

We thank the Reviewer for her/his favorable comments.

Reviewer 3 Report

The current review exhibits a brilliant consolidation of various medical complications arising due to diabetes mellitus and their underlying inflammatory mechanism. The authors gave good insight into the contribution of the inflammasome in the pathogenesis of various metabolic conditions arising due to diabetes mellitus. Such a precise focus on the role of NLRP3 inflammasome makes them the most significant therapeutic target. The authors also brilliantly consolidated many areas that have the potential for future research. Such updated information opens up wide therapeutic and treatment options for numerous chronic complications.  The present review meets the quality of scientific writing and can be improved after revision.

Major Revisions:

Comment 1:

The authors explained the role of lincRNA Gm4419 in the activation of NLRP3 inflammasome in glomerular mesangial cells exposed to high glucose condition. What is the role of lncRNAs in other diabetes mellitus related metabolic diseases? The authors are encouraged to briefly describe. A detailed understanding of the mechanism underlying NLRP3 inflammasome activation in various Diabetic Mellitus related complications would help the readers to understand the subject better.

Comment 2:

Are there any natural anti-diabetic medicines available? The authors are recommended to describe the existing natural anti-diabetic medicines and how they regulate the NLRP3 inflammasome in the case of Diabetic Mellitus.

Minor Revision:

Comment 1:

The authors are suggested to proofread the manuscript to avoid minor mistakes.

Author Response

Reviewer #3

The current review exhibits a brilliant consolidation of various medical complications arising due to diabetes mellitus and their underlying inflammatory mechanism. The authors gave good insight into the contribution of the inflammasome in the pathogenesis of various metabolic conditions arising due to diabetes mellitus. Such a precise focus on the role of NLRP3 inflammasome makes them the most significant therapeutic target. The authors also brilliantly consolidated many areas that have the potential for future research. Such updated information opens up wide therapeutic and treatment options for numerous chronic complications.  The present review meets the quality of scientific writing and can be improved after revision.

We thank the reviewer for her/his overall positive judgement and valuable suggestions and feedback in improving our manuscript.

  1. The authors explained the role of lincRNA Gm4419 in the activation of NLRP3 inflammasome in glomerular mesangial cells exposed to high glucose condition. What is the role of lncRNAs in other diabetes mellitus related metabolic diseases? The authors are encouraged to briefly describe. A detailed understanding of the mechanism underlying NLRP3 inflammasome activation in various Diabetic Mellitus related complications would help the readers to understand the subject better.

Following to the Reviewer suggestion, we have now added a brief comment stating that “recently, there has been an increasing interest in the ncRNA field and, specifically, in the role of ncRNAs in chronic complications of diabetes and related metabolic disorders”.  However, since this topic is not within the scope of the review and, apart from the article by Yi et al (see ref #104 of the revised manuscript), there are no studies investigating the crosstalk between lncRNAs and the inflammasomes in diabetic complications, we referred the interested readers to the most recent literature on the topic of lncRNAs in diabetic complications unrelated to inflammasome (see refs #105-107 of the revised manuscript].

  1. Are there any natural anti-diabetic medicines available? The authors are recommended to describe the existing natural anti-diabetic medicines and how they regulate the NLRP3 inflammasome in the case of Diabetic Mellitus.

We share with the Reviewer the interest on natural compounds as potential therapeutic strategies for diabetes and its complications. However, in our opinion, this is a vast field that deserves a separate review article. Therefore, we commented this important topic of research and referred the interested readers to the most recent literature on phytochemicals as regulators of NLRP3 activity in experimental models of diabetes mellitus (se ref #199 of the revised manuscript).

Minor revision

The authors are suggested to proofread the manuscript to avoid minor mistakes.

As suggested, the manuscript has been carefully checked for spelling errors, continuity errors, punctuation errors, grammar errors, etc.